# Classification of Virtual Harassment on Social Networks Using Ensemble Learning Techniques

Nureni Ayofe Azeez [1,*] and Emad Fadhal [2,*]

1    Department of Computer Sciences, University of Lagos, Lagos 100213, Nigeria
2    Department of Mathematics and Statistics, College of Science, King Faisal University, P.O. Box 400, Al Hofuf 31982, Al-Ahsa, Saudi Arabia
*    Correspondence: nazeez@unilag.edu.ng (N.A.A.); efadhal@kfu.edu.sa (E.F.)

**Abstract: Background:** Internet social media platforms have become quite popular, enabling a wide range of online users to stay in touch with their friends and relatives wherever they are at any time. This has led to a significant increase in virtual crime from the inception of these platforms to the present day. Users are harassed online when confidential information about them is stolen, or when another user posts insulting or offensive comments about them. This has posed a significant threat to online social media users, both mentally and psychologically. **Methods:** This research compares traditional classifiers and ensemble learning in classifying virtual harassment in online social media networks by using both models with four different datasets: seven machine learning algorithms (Nave Bayes NB, Decision Tree DT, K Nearest Neighbor KNN, Logistics Regression LR, Neural Network NN, Quadratic Discriminant Analysis QDA, and Support Vector Machine SVM) and four ensemble learning models (Ada Boosting, Gradient Boosting, Random Forest, and Max Voting). Finally, we compared our results using twelve evaluation metrics, namely: Accuracy, Precision, Recall, F1-measure, Specificity, Matthew's Correlation Coefficient (MCC), Cohen's Kappa Coefficient KAPPA, Area Under Curve (AUC), False Discovery Rate (FDR), False Negative Rate (FNR), False Positive Rate (FPR), and Negative Predictive Value (NPV) were used to show the validity of our algorithms. **Results:** At the end of the experiments, For Dataset 1, Logistics Regression had the highest accuracy of 0.6923 for machine learning algorithms, while Max Voting Ensemble had the highest accuracy of 0.7047. For dataset 2, K-Nearest Neighbor, Support Vector Machine, and Logistics Regression all had the same highest accuracy of 0.8769 in the machine learning algorithm, while Random Forest and Gradient Boosting Ensemble both had the highest accuracy of 0.8779. For dataset 3, the Support Vector Machine had the highest accuracy of 0.9243 for the machine learning algorithms, while the Random Forest ensemble had the highest accuracy of 0.9258. For dataset 4, the Support Vector Machine and Logistics Regression both had 0.8383, while the Max voting ensemble obtained an accuracy of 0.8280. A bar chart was used to represent our results, showing the minimum, maximum, and quartile ranges. **Conclusions:** Undoubtedly, this technique has assisted in no small measure in comparing the selected machine learning algorithms as well as the ensemble for detecting and exposing various forms of cyber harassment in cyberspace. Finally, the best and weakest algorithms were revealed.

**Keywords:** harassment; classification; ensemble; metrics; algorithm; learning; classifiers





## 1. Introduction

Machine learning, which is a subset of Artificial Intelligence, is the study of data-driven methods capable of imitating, understanding, detecting, and assisting human and genetic information processing tasks. Many related issues arise, such as how to collate, sort, compress, interpret, and process data. Often, these methods are not necessarily directed to imitating human processing directly, but rather to enhance data processing, such as in predicting the outcome of an event rapidly. This agent includes software that is considered

an "Intelligent System". It perceives its environment and takes necessary actions to perform the task for which it was designed, irrespective of the circumstance it faces. All of these functions are carried out autonomously, i.e., taking decisions, thinking, etc. Retrieval of data sets from a repository such as Bayzick, Kaggle, Instagram, Facebook, and YouTube, respectively, can be carried out manually. Based on the huge volume of data available, there is a need to apply machine learning algorithms that help classify these datasets appropriately. The algorithms include Naive Bayes', Random Forest, KNN (K-nearest Neighbour), Support Vector Machine, and many more.

Cyberspace is undoubtedly characterized with social challenges among internet users, specifically among the youth [1]. They tend to mismanage the internet for carrying out several activities that may affect the personality of others. Some of the social challenges which an individual can suffer from when using the internet are: phishing, virtual harassment, cyberbullying, and cyberstalking.

Having realized the potential damage this scenario might cause [2], this research aims to classify any form of harassment in the cyberspace context using an ensemble learning approach.

These algorithms have advanced the analysis of virtual harassment as part of the existence of vulnerability in the structure, design, and architecture of computers and other smart electronic and computational devices, especially in the operating system, applications, and network structure. The identification and evaluation of the network node vulnerability in the case of virtual harassment is a key issue in information security research and this has attracted more attention in recent times. Due to the availability of various algorithms for classifying data, there's a need for measuring and comparing the performance of each algorithm with another to choose the best algorithm for a specific task which results in an accurate result from data classification for better management decisions, in this case, using these algorithms in virtual harassment detection.

### 1.1. Research Limitation

This research focuses on detecting virtual harassment on social media networks with four datasets using machine learning algorithms and ensemble learning. Though there are still several other social media vices in cyberspace apart from virtual harassment, adequate consideration is given to this in order to have a comprehensive overview and analyzable results. Indeed, the importance of this study cannot be overemphasized, as it is important and beneficial to individuals, educational institutions, and corporate entities that are keen on making profitable decisions in their daily business transactions on the internet. Limiting the research focus to this has undoubtedly assisted in providing useful information on the subject matter and appropriate insight into similar future research.

### 1.2. Organization

The remaining part of the paper is organized as follows: Section 2 provides background information on the research work; Section 3 presents the related works; Section 4 summarizes the data collected and their corresponding sources; Section 5 presents the discussion of results; Future work is presented in Section 6; and the conclusion is in Section 7.

### 2. Background

Nearly all the countries of the world are relying solely on the applications of ICT in running their day-to-day activities. As a matter of fact, some cannot execute any financial obligations or govern their daily procedures without connecting to cyberspace because of the accuracy, timeliness, and real-time benefits that it provides. Youth across the globe cannot be left out of the usage of this ubiquitous technological initiative. They research, communicate, and share their views on issues with this innovation seamlessly and at little or no cost. Having noticed some of the main benefits of ICT, it has also been established that some noticeable challenges are circumventing the full realization and optimization of its benefits. One of the major challenges hampering the utilization of

this technology is global cyber-harassment. It has affected some fundamental benefits that individuals, countries, and the world as a whole could gain. Virtual harassment is a new form of harassment that follows both students and working adults from the perimeters of their various schools and offices directly into their homes. It is a current major problem among both old and young people (teenagers included) who utilize social media platforms for communication and information sharing. The vast majority of victims of virtual harassment are being bullied from the moment they wake up to check their social media accounts and handles on their phone or email until they shut down their computers, laptops, phones, or any other computational device. This has had a significant negative impact on youth and teenagers' emotions, psychological well-being, intimidation, low self-esteem, and interpersonal relationships. This is because the so-called harasser can be anywhere, victimizing another person from the comfort of his own home. The need to take the necessary action to help reduce and detect virtual harassment posts on social media [3] platforms is very urgent and important. Against this backdrop, this research aims at evaluating ensemble learning models and traditional machine learning algorithms for detecting virtual harassment with four (4) different datasets obtained from reliable sources. The intention is to compare and evaluate machine learning algorithms and ensemble learning models for detecting virtual harassment, present the results, and conclude how to detect virtual harassment on online social platforms.

## 3. Related Works

According to [4], online harassment is referred to as "virtual harassment." The use of email, instant messaging, and offensive websites to bully or otherwise harass a person or group are known as "virtual harassment". Flames, remarks made in chat rooms, the sending of rude or nasty emails, or even disturbing others by commenting on blogs or social networking sites are all examples of virtual harassment.

The Oxford Dictionary states, "Virtual harassment can be defined as the use of electronic communication to bully a person, typically by sending messages of an intimidating or threatening nature" [5].

According to Dictiory.com, "Virtual harassment can be defined as the act of harassing someone online by sending or posting mean messages, usually anonymously" [6].

According to the Cambridge Dictionary, "Virtual harassment can be defined as the activity of using the internet to harm or frighten another person, especially by sending them unpleasant messages" [7].

According to Merriam-Webster Dictionary, "Virtual harassment is the electronic posting of mean-spirited messages about a person (such as a student) often done anonymously" [8,9]. Virtual harassment includes body shaming, religious discrimination, name-calling, and racism in cyberspace [10].

Numerous techniques have been put out in recent years to measure, detect, and assess offensive or degrading social media content and behavior [11].

Nureni et al., 2021 Used Twitter datasets to analyse well-known classification techniques and to suggest an ensemble model for detecting instances of cyberbullying. Naive Bayes, KT Nearest Neighbours, Logistic Regression, Decision Tree, Random Forest, Linear Support Vector Classifier, Adaptive Boosting, Stochastic Gradient Descent, and Bagging classifiers are some of the techniques used for the evaluation [12]. In experiments, the classifiers were compared against four metrics: accuracy, precision, recall, and F1 score. The outcomes show how each algorithm's performance compared to its relevant measures. Compared to the linear support vector classifier (SVC), the ensemble model produced better results of all. The medians for the Random Forest classifier across the datasets are 0.77, 0.73, and 0.94, making it the top-performing classifier. With medians of 0.77, 0.66, and 0.94 compared to the linear support vector classifier's 0.59, 0.42, and 0.86, the ensemble model has demonstrated an improvement in the performance of its constituent classifiers [13].

Manuel et al., 2021 Followed two supervised learning methods, namely: threshold and dual. Results showed how to improve baseline detection models by up to 42%. Experiments with the dataset from some other social media platforms used Random Forest for negative models and an Extra tree for the positive models [14].

Furthermore, Celestine et al., in 2020, conducted an empirical analysis to determine the effectiveness and performance of deep learning algorithms in detecting insults in social media commentary. Results showed that (Bidirectional Long Short-Term memory) BLSTM model achieved high accuracy and F1-measure scores in comparison to (Recurrent Neural Network) RNN, (Long Short-Term Memory) LSTM, and (Gated Recurrent Units) GRU [15].

Many methods have been proposed in the past year to quantify and detect unpleasant or insulting content and behaviours on Instagram, YouTube, 4chan, Yahoo Finance, and Yahoo Answers [16].

Chen et al., 2012 combined physical and written characteristics (such as the ratio of imperative sentences, verbs, adverbs, and adjectives as offensive words) [16] to foresee a user's ability in creating aggressive content [17] in the comments made on YouTube, while Djuric et al. depended on word implanting to distinguish foul comments on Yahoo Finance. Nobata et al. implemented hate speech recognition on Yahoo Finance and News Data, using a classification based on supervised learning. Kayes et al., 2015 discovered that users have a habit of flagging offensive content forwarded [2,16,18–20].

Dinakar et al. detected virtual harassment by disintegrating it into the detection of sensitive topics. Comments from YouTube were collected from contentious videos using manual comments to distinguish them and perform a "bag-of-words" driven text classification. Van Hee et al. learnt language features in virtual harassment-related content pulled out from Ask.fm, intending to detect fine-grained types of virtual harassment, such as intimidations and abuses. In addition to the target and harasser, they also identified bystander protectors and eyewitness assistants, who support, individually, the victim or the harasser [21,22].

Hosseinmardi et al., 2015 studied pictures posted on Instagram and their related comments to detect and distinguish between virtual aggression and virtual harassment. In conclusion, authors (Saravanaraj et al., 2016) offered an attitude for detecting harassment arguments found in Twitter [23] tweets, as well as demographics about bullies (such as their age and gender) [24].

A comparable graph-based approach is also used by Hosseinmardi et al., 2015. The study of a text from the perspective of Romanticism can also add useful structures in detecting aggressive or insulting content. For instance, Nahar et al., 2012 used sentiment records of data collected from Kongregate (an online gaming site) [25], Slashdot, and MySpace [26,27].

Nandhini & Sheeba, 2015 proposed a model that uses the Naïve Bayes machine learning approach [27–36]. They attained 91% accuracy in their dataset, which was retrieved from MySpace.com [37], and then they projected an additional model [38]. Naïve Bayes Classifier and Genetic Operations (FuzGen) attained 87% accuracy. Another approach, by Walisa, Lodchakorn, Pimpaka, Piyaporn, and Pirom, 2017 improved the Naïve Bayes Classifier for removing the words and investigating loaded pattern gathering [39–41]. Using this approach, they achieved 95.79% accuracy on datasets from Slashdot, Kongregate, and MySpace [42]. Nevertheless, they had difficulties with the clustering process because it does not work in a parallel manner [43]. Likewise, in the methodology proposed by Shane et al., 2018 the War of Tanks chat was used to collect their dataset and classified them manually. Comparisons were made with the simple Naïve Classification that makes use of emotional exploration as a characteristic [44]. They had poor computation results when the dataset classified manually was compared with theirs [45,46].

Furthermore, Saravanaraj et al., 2016 proposed a method using their dataset from Kaggle that employed two different classifiers, namely: Naïve Bayes and SVM [24]. The Naïve Bayes Classifier produced an average correctness of 92.81%, while SVM with poly kernel yielded an accuracy of 97.11%; however, they did not reference the size of the dataset

used for testing. There is the possibility that their result might not be trustworthy [25,47–49]. Another Approach by Karthik et al., 2012 intended to distinguish clear harassment language relating to (1) sexuality (2) race and Culture, and (3) intelligence. The data set used was retrieved from YouTube comments. Two different classifiers were used to generate the results, namely, SVM and Naïve Bayes. SVM produced a correctness of 66%, while Naïve Bayes produced a correctness of 63% [50].

Michele et al., 2016 projected a new method for detecting virtual harassment by implementing an unsupervised approach. They used the classifiers inconsistently over their dataset, using SVM on FormSpring and achieving 67% on the ability to remember, applying GHSOM on YouTube. Their results yielded 60% exactness, 69% correctness, and 94% remembrance. Applying Naïve Bayes on Twitter, they attained 67% correctness [51]. Furthermore, Batoul et al., 2017 came up with a model to detect virtual harassment carried out in Arabic. They used Naïve Bayes and achieved an accuracy of 90.85%. With SVM, they achieved 94.1% exactness, but the rate of false-positive was very high [52].

Another type of method using "Deep Learning and Neural Networks" is in the paper by Xiang, et al., 2016. They used novel enunciation centred on a sophisticated neural network, thereby lessening the difficulty related to noise and Harassment data scarcity to the counter imbalance in the class [53]. They retrieved 1313 messages from Twitter and 13,000 messages from Formspring.me [54]. They were unable to calculate the accuracy of the dataset retrieved from Twitter because they were imbalanced. They achieved 56% exactness, 78% recall, and 96% accuracy. Even though they achieved high accuracy, their dataset was unbalanced, therefore, producing incorrect output reflected in the score of exactness of 56%. Chikashi, Joel, Achint, Yashar, and YI, 2016 showed the recent increase in abusive language using a framework called Vowpalwabb. They also established a supervised classification methodology with NLP structures that outclassed the deep learning approach, The F-Score extended to 0.817 using a dataset retrieved from Yahoo News and Finance comments [55].

In [56], the authors proposed a classification technique that depends on fusing information that is captured with images that show the same object from multiple angles. They adopted convolutional neural networks, which were utilized for the extraction and encoding of various visual features [45,57–60].

In the work of [61], an attempt was made to study touch-based gestures to distinguish between adults and children who might be accessing a smartphone and to ensure protection [62,63]. They adopted machine learning algorithms alongside a developed Android app to evaluate the so-called techno-regulatory approach, which has 9000 touch gestures from almost 150 respondents.

A comprehensive review of the basic theories for knowing the features and characteristics of multi-view learning is provided in [64,65]. A standard taxonomy was provided based on machine learning techniques as well as the styles in which diverse views are utilized and exploited. The main objective of the review was to provide insight into the current happenings in the multi-view learning field [66].

A novel technique was proposed for providing security awareness to the internet user using a sentence-embedding approach and machine learning in order to have absolute control over their classified information while using the internet. The technique proposes four modules for realizing its objectives. The modules, which are the keyword module, the topic module, the sensitiveness module, and the personalization module, attain their effectiveness in terms of sensitive information protection and identification, as well as their efficiency in terms of impact on user application [67].

PrivScore, which is a context-aware, text-based quantitative model for personal classified information assessment, was proposed by [68]. The motive behind this technique is to provide a platform for alerting individuals to possible information leakage. The authors solicited various opinions on the sensitive nature of private information from crowdsourcing workers and analyzed the feedback to understand the perceptual model behind the disagreements and agreements.

Over the years, some of the solutions provided in solving various forms of vice on the internet have been mostly curative but not preventive measures. In an attempt to institute a preventive measure for solving cyberbullying, Prabhu's patented research in 2015 tagged "Method to stop cyberbullying before it occurs". The ReThink App was designed to verify and determine the hurtful nature of a text, video, or image before posting it on the internet. This app has undoubtedly proven to be very useful and effective among adolescent internet users [69].

Rui et al., 2016 suggested a framework that was mainly used for detecting virtual harassment. They made use of embedded words that are similar to insulting words. Weights were assigned to those words to obtain the features related to harassment. SVM was used as the main classifier and obtained correctness of 79.4% [59,70]. Sourabh and Vaibhav, 2014 projected an extra method. They collected their dataset from MySpace and marked them manually, and then they used a Support Vector Machine classifier for their classification [71,72]. The summary of the reviewed papers is hereby presented in Table 1.

**Table 1.** Summary of related articles.

| Author | Year | Approach | Strength | Weakness |
|---|---|---|---|---|
| (Nureni, Sunday, Chinazo, & Charles) [12] | 2021 | Naive Bayes, K-Nearest Neighbors, Logistic Regression, Decision Tree, Random Forest, Linear Support Vector Classifier, Adaptive Boosting, Stochastic Gradient Descent, and Bagging classifiers are some of the techniques used for evaluation. | The medians for the Random Forest classifier across the datasets are 0.77, 0.73, and 0.94, making it the top-performing classifier. With medians of 0.77, 0.66, and 0.94 compared to the linear support vector classifier's 0.59, 0.42, and 0.86. | The ensemble model has demonstrated an improvement in the performance of its constituent classifiers |
| (Manuel, Francisco, Victor, & Fidel) [14] | 2021 | Followed two supervised learning methods namely:<br>1. Threshold<br>2. dual | Results show how to improve baseline detection models by up to 42% | Experiment with a dataset from some other social media platforms.<br>Use random forest for negative models.<br>Extra tree for the positive models. |
| (Celestine, Gautam, Suleman, & Praveen) [15] | 2020 | Empirical analysis to determine the effectiveness and performance of deep learning algorithms in detecting insults in social media commentary. | Results show that the BLSTM model achieved high accuracy and F1-measure scores in comparison to RNN, LSTM, and GRU. | Deep learning models can be most effective against cyberbullying when directly compared with others and paves the way for future hybrid technologies that may be employed to combat this serious online issue. |
| (Abaido) | 2019 | Enhance Timing Approach (ETA) and Ensemble learning. | SPSS was used for the reliability test and it showed satisfactory results for the research study (Alpha = 0.718) further results showed that virtual harassment exists on social media platforms at 91% positive. | Further quantitative research is required to assess the socio-psychological impacts of virtual harassment on victims in conservative societies |
| (Shane, William, Adrian, & Gordon) [45] | 2018 | Datasets were gotten from the war of Tanks game and classifications were done manually. | It has a similarity with the Simple Naïve classification that uses emotional analysis. | The results produces were very poor |

**Table 1.** *Cont.*

| Author | Year | Approach | Strength | Weakness |
|---|---|---|---|---|
| (Walisa, Lodchakorn, Pimpaka, Piyaporn, & Pirom) [22] | 2017 | Improved Naïve Bayes classifier was used to eliminate words and examine the loaded pattern | 95.79% correctness was achieved after the experiment | The cluster pattern does not work in parallel |
| (Sani & Livia) [60] | 2017 | Two classifiers were used Naïve Bayes and SVM and the data set was collected from Kaggle | 92,81% accuracy for Naïve Bayes and 97.11% for SVM | The dataset used for testing and training was not mentioned, hence their result isn't credible |
| (Batoul et al.) [52] | 2017 | They made use of the Arabic language and the classifiers used were Naïve Bayes and SVM | 90.85% precision with Naïve Bayes and 94.1% precision on SVM | The result had a high rate of false Positive |
| (Michele, Emmanuel, & Alfredo ) [51] | 2016 | An unsupervised learning approach was used | Accuracy of 67%, 60%,69%, 94% and 67% were achieved | The average levels of accuracy were low than when compared with supervised learning algorithms |
| (Celestine, et al.) [15] | 2016 | Deep learning and Neural Networks approaches were used for the experiment | 56% exactness, 70% recall, and accuracy 96% | The data set was unbalanced while achieving high accuracy, so it gave incorrect output |
| (Rui, Anna, & Kezhi) [59] | 2016 | Word embedding makes a list of pre-defined words | 79.4% accuracy using Support Vector Machine | Only one classifier was used |
| (Chikashi, Joel, Achint, Yashar, & YI) [55] | 2016 | Vowpalwabbit framework was used for classification and NLP features | It performs better when compared with the deep learning approach with about 81% accuracy | the other classifiers such as Naive Bayes gave better accuracy |
| (Nandhini & Sheeba) [73] | 2015 | Naïve Bayes machine learning Effort | 91% Accuracy was achieved | Efficiency is reduced when tried with another classifier |

*Methodology*

The application of traditional machine learning algorithms (Decision Tree algorithm –, K-Nearest Neighbor (KNN) algorithm, Logistics Regression algorithm, Naïve Bayes, Neural Network (NN), Quadratic Discriminant Analysis, Support Vector Machine (SVM), and ensemble learning) was adopted. Application of four different datasets as presented in Table 2. Data Pre-processing, which is an essential part of text classification, was carried out on the datasets because they contained a large amount of vague information that needed to be eliminated.

**Table 2.** Dataset source.

| Dataset | URL | Source | Remark |
|---|---|---|---|
| Dataset 1 | https://github.com/jo5hxxvii/cyberbullying-text-classification | Kaggle | GitHub, 2022 |
| Dataset 2 | https://github.com/jo5hxxvii/youtube_parsed_dataset-text-classification | You-tube | GitHub, 2022 |
| Dataset 3 | https://www.bayzick.com/bullying_dataset | Bayzick | Bayzick.com |
| Dataset 4 | https://github.com/jo5hxxvii/aggression_parsed_dataset-text-classification | Kaggle | GitHub, 2022 |

## 4. Results

### 4.1. Data Collection

#### 4.1.1. Dataset 1

The First dataset was extracted from Kaggle to extensively make a study among LA and Boston youths who are between 18–40 years. The Kaggle-Parsed dataset consisted of a total of 8800 (eight thousand, eight hundred) tokens that were divided into chats where the youths raised concerns about the presidency and 9/11. The dataset was collected from the GitHub website. The extracted zip file dataset for aggression contained extensive information publicly available, such as their username and their public post—\protect\unhbox\voidb@x\hbox{https://github.com}/jo5hxxvii/cyberbullying-text-classification. Accessed on 13 January 2023.

#### 4.1.2. Dataset 2

The Second dataset was retrieved from YouTube; https://github.com/jo5hxxvii/youtube_parsed_dataset-text-classification which includes a large number of YouTube comment pages publicly made available on the site regarding the topic of random shootings by teenagers and young adults between the age of 14 and 56 years. The dataset contained text that may be considered vulgar, abusive, disrespectful, threatening, and suicidal. The dataset was used to check toxic comment classification challenges.

#### 4.1.3. Dataset 3

Dataset 3 was also collected from the Bayzick website—https://www.bayzick.com/bullying_dataset (Accessed on 13 January 2023) and it can be accessed via "cyberbullying-text-classification-main/BayzickBullyingData/HumanConcensus". Dataset 3 is another set of a large number of Twitter comments, which contained toxic behaviours, sexism, and racism.

#### 4.1.4. Dataset 4

The Second dataset was collected from the Kaggle website—https://github.com/jo5hxxvii/aggression_parsed_dataset-text-classification (Accessed on 13 January 2023) which included a large number of 115,863 (one hundred and fifteen thousand, eight hundred and sixty-three) comments. Rabbinic/pharisaic Judaism comments made publicly available on the site made improvements to the current model to help online chats become more productive and respectful. The dataset contained text that may be considered religious biases, tribalism, and racism.

### 4.2. Machine Learning Algorithms

In this research work, seven different algorithms (classifiers) and four different ensemble learning methods were used in this research, along with training the data set in the course of the implementation. Those machine learning algorithms include:

#### 4.2.1. Decision Tree Algorithm

Decision trees are easier to understand than their random forest counterpart, which synthesizes numerous decision trees into a single model and may be more effective for multiclass classification and other challenging artificial intelligence problems. They belong to the family of supervised learning algorithms. Problems are solved using tree representations. Each internal node of the tree corresponds to an attribute and each leaf node corresponds to a class label. To put it simply, decision trees perform best in straightforward situations with few variables, but neural networks excel in situations where the data includes complicated correlations between characteristics or values (i.e., is "dense"). The classification or decision is represented by the leaf node while the node has two or more branches.

$$E(S) = \sum_{i=1}^{c} -p_1 log_2 p_1 \tag{1}$$

where $p$ is the $i$-th order probability,

$$G(S,C) = E(S) - \sum_{w \in values(C)} \frac{S_w}{S} E(S_w) \tag{2}$$

### 4.2.2. KNN

One of the simplest categorization techniques, a parametric classification method, is the K-nearest neighbours approach. It compares every example of a given class and every other example of that class. This is simple to use, comprehensive, and remarkably accurate. An object is categorized based on the majority vote of its neighbours and is then put into the class with the highest percentage of support among its K-closest neighbours. (K is a user-defined constant that is often a small, positive integer.) Data points are converted into feature vectors to execute KNN, which has its foundation in mathematical theories, or their arithmetic equivalent

$$d(p,q) = d(q,p) = \sqrt{(q_1 - p_1)^2 + (q_2 - p_2)^2 + \cdots + (q_n - p_n)^2} \tag{3}$$

$$= \sqrt{\sum_{i=1}^{n}(qi - Pi)^2} \tag{4}$$

where $q_1$ to $q_n$ represents the attribute's value for one observation and $p_1$ to $p_n$ represents the attribute value for the other observation.

### 4.2.3. Logistics Regression

We can manage classes by using logistic regression, which is a non-linear extension of linear regression. This is accomplished by categorizing predictions according to a probability threshold. Logistic regression is frequently utilized in practical applications, including multi-label classification difficulties and estimating creditworthiness across a range of categories. The Naive Bayes classifier, which applies Bayes' Theorem and produces a larger bias but lower variance, can be replaced with logistic regression.

- We know the equation of the straight line can be written as:

$$y = b_0 + b_1 x_1 + b_2 x_2 + b_3 x_3 + \cdots + b_n x_n \tag{5}$$

- In Logistic Regression $y$ can be between 0 and 1 only, so we divide the above equation by $(1 - y)$:

$$\frac{y}{1-y}; 0 \ for \ y = 0, and \ infinity \ for \ y = 1 \tag{6}$$

- We need a range between $-$[infinity] to $+$[infinity], then, if we take the logarithm of the equation, it will become:

$$log\left[\frac{y}{1-y}\right] = b_0 + b_1 x_1 + b_2 x_2 + b_3 x_3 + \cdots + b_n x_n \tag{7}$$

The above equation is the final equation for Logistic Regression.

The sigmoid function is an S-shaped curve that can acquire any real-valued number and map it onto a value between the range of 0 and 1, but never exactly those limits.

$$1/\left(1 + e\hat{} - value\right) \tag{8}$$

where $e$ is equal to the base of the natural logarithm (Euler's number) and *value* is equal to the actual numerical value to be transformed [38].

### 4.2.4. Naïve Bayes

Naive Bayes classifiers are a subset of linear classifiers that assume that the value of a particular feature is independent of the value of any other feature. This means that we can use Bayes' Theorem to calculate the probability of a particular label given to our data by

just looking at each feature individually, without considering how features may interact with each other. Naive Bayes classifiers are often used in text classification because they easily calculate probabilities from frequencies, and text typically has a large number of features (e.g., individual tokens in words). They are also popular in spam detection because they can deal with the high dimensionality of email data (e.g., all the different words used in an email) without overfitting the data.

$$p\left(X_j\middle|Y =_{yk}\right) = \theta_{kj}^{X_j}\left(1 - \theta_{kj}\right)^{1-X_j} \tag{9}$$

In addition, the Bernoulli Distribution is an independent probability function in which a random variable can take one of two potential values—1 for success or 0 for failure.

Given a class variable or hypothesis ($y$) and a dependent feature or evidence ($x1$–$xn$),

$$P(y|x1, x2, x3 \dots xn) = \frac{P(y)P(x1, x, x3, \dots \dots xn|y)}{P(x1, x, x3, \dots \dots xn)} \tag{10}$$

where: $P(y)$ are labels

$P(x)$ are comments

$P(y|x1, x2, x3 \dots xn)$ is how probable was the hypothesis (labels) given the observed evidence (comments)?

$P(x1, x2, x3 \dots xn|y)$ is how probable is the evidence, given that the hypothesis is true?

$P(y)$ is how probable was the hypothesis before observing the evidence?

$P(x1, x2, x3 \dots \dots xn)$ is how probable is the new evidence under all possible hypotheses?

### 4.2.5. Neural Network

As their name implies, they simulate biological brain networks in computers. More parameters are needed for more sophisticated models, which might make them slower than simpler techniques, such as logistic or linear regression algorithms, in classifying new data points [74]. However, because of their adaptability and scalability, they can easily handle enormous amounts of unlabelled data. A collection of parameters is trained on data using ANNs. The model's output, which could be either an input or an action, is then determined using these parameters. The neurons that make up each layer of the network generally correlate to particular characteristics or qualities. The perceptron is the most basic type of artificial neural network [75].

The neural network equation is the following:

$$Z = Bias + W_1 X_1 + W_2 X_2 + \dots + WnXn \tag{11}$$

where,

- $Z$ is the symbol for denotation of the above graphical representation of ANN;
- $W$ is, are the weights or the beta coefficients;
- $X$ is, are the independent variables or the inputs;
- *Bias* or intercept = $W_0$

### 4.2.6. Quadratic Discriminant Analysis

The LDA variation, known as Quadratic Discriminant Analysis (QDA), enables non-linear data separation. This is accomplished by using a quadratic curve as opposed to a linear boundary to match your data. Due to the quadratic operation required to determine the within-class variance for each class, QDA is more computationally demanding. However, if you have a large amount of training data and you think that the classes in your data are not linearly separable, QDA might be a better option than LDA.

$$x = \frac{-b \pm \sqrt{b^2 - 4ac}}{2a} \tag{12}$$

### 4.2.7. Support Vector Machine

Support Vector Machines (SVM), which convert your data into a linear decision space, are reliable and efficient machine learning methods. After that, the algorithm chooses the best hyperplane in this linear decision space to divide your training data into distinct classes, such as valid and invalid emails. SVM excels at distinguishing related objects. Support Finding the "hyperplane we $x = 0$" that maximizes the margin between the two classes—which may be done by solving a quadratic objective function—allows Vector Machine to distinguish between positively and negatively labelled data [76].

$$h(x_1) = \begin{cases} +1 \ if \ w.x + b \geq 0 \\ +1 \ if \ w.x + b < 0 \end{cases} \tag{13}$$

### 4.3. Ensemble Learning

An ensemble method or ensemble learning algorithm consists of aggregating multiple outputs made by a diverse set of predictors to obtain better results. Formally, based on a set of "weak" learners, we are trying to use a "strong" learner for our model. The suggested ensemble model is a method that combines various machine learning classifiers and models to outperform the individual models. On the dataset, each component classifier is trained to make predictions. A final prediction is then created by combining these predictions. There are several ways to reach this conclusion, including stacking, voting, bagging, and boosting. Voting is employed in this study to determine the outcome. Here, the majority rule is implemented through the use of projected class labels for voting. The ensemble uses the constituent estimators' Multinomial NB, Linear SVC, and Logistic Regression [13].

### 4.4. Experiments

Extensive experiments were run to measure the performance of the classifiers (Decision Tree DT, K-Nearest Neighbor KNN, Logistics Regression LR, Bernoulli Naïve Bayes NB, Neural Network NN, Quadratic Discriminant Analysis QDA, and Support vector machine SVM) used during the project. Furthermore, extensive experiments were also run to measure the performance of our Ensemble learning models (Ada Boosting, Gradient Boosting, Random Forest, and Max Voting) in detecting virtual harassment.

### 4.5. Performance Metrics

To evaluate the performance of the traditional classifiers and ensemble learning on the test data where the true values are known, a confusion matrix was used. The performance measures considered in this project include accuracy, recall, precision, and the F1 score, which is determined from the confusion matrix.

- True Positive (TP): This instance indicates virtual harassment that was classified as virtual harassment;
- True Negative (TN): This instance indicates non-virtual harassment samples that were classified as non-virtual harassment;
- False Positive (FP): This instance indicates virtual harassment samples that were classified as non-virtual harassment;
- False Negative: It indicates non-virtual harassment samples that were classified as virtual harassment.

Performance metrics such as accuracy, sensitivity, and specificity are the most widely used in medicine and biology.

The performance metrics are presented mathematically below:

$$Accuracy = \frac{True \ Positive + True \ Negative}{Total \ Example} \tag{14}$$

$$Recall = \frac{True \ Positives}{True \ Positive + False \ negatives} \tag{15}$$

$$Precision = \frac{True\ Positive}{True\ Positive + False\ Positives} \tag{16}$$

$$F1 = 2 \times \frac{Precision \times Recall}{Precision + Recall} \tag{17}$$

The labelled texts are vectorized which helps to extract features or tokens from each text and a numerical value is assigned to each word, token, or feature in the text. Then test datasets are generated using the 70 per cent of the dataset available used to train the classifiers, while 30 per cent is used to test the classifiers. The output of each classifier is displayed in a confusion matrix, where you have True Positive (labelled as positive and predicted as positive), True Negative (labelled as negative and predicted as negative), False Positive (labelled as negative but predicted as positive), and False Negative (labelled as positive but predicted as negative) as well as performance parameters, such as Accuracy, Precision, Recall, F-measure, specificity, MCC, KAPPA, AUC-Area Under Curve, FDR-False Discovery Rate, FNR-False Negative Rate, FPR-False Positive Rate, and NPV-Negative Predictive Value, are calculated from the confusion matrix. The confusion matrix, which is a method of succinctly presenting the performance and efficiency of classification algorithms.is hereby presented in Table 3.

**Table 3.** The confusion matrix.

|  | **PREDICTED POSITIVE** | **PREDICTED NEGATIVE** |
|---|---|---|
| **ACTUAL POSITIVE** | TRUE POSITIVE (TP) | FALSE NEGATIVE (FN) |
| **ACTUAL NEGATIVE** | TRUE NEGATIVE (TN) | FALSE POSITIVE (FP) |

## 5. Discussion

### 5.1. Data Analysis for Dataset 1

The seven classifiers used in the course of this project include Decision Trees, K-nearest Neighbor, Logistics Regression, Naïve Bayes, Neural Network, Quadratic Discriminant Analysis, Support vector machine (SVM) as well as four Ensemble learnings which are: Ada Boost, Gradient Boosting, Random Forest algorithms, and Max voting were extensively trained using our dataset and outputs were gotten from each of the algorithms. Each of the algorithms considering our performance metrics, as stated in chapter three, produced an output result consisting of Accuracy, Precision, Recall, Specificity, MCC, KAPPA, F1 Score, AUC, FDR, FNR, FPR, and NPV. The accuracy levels help us know which of the classifiers performs best when it comes to detecting virtual harassment.

Table 4 shows the result of the machine learning algorithm and ensemble learning for Dataset 1 [50]. Logistics Regression has the highest accuracy score of 0.6923, followed by the Support Vector Machine and Neural Network, which both attained an accuracy of 0.6897 and 0.6827, respectively. Decision Trees followed, with an accuracy of 0.6487, then K Nearest Neighbor, KNN, and Quadratic Discriminant Analysis. QDA had an accuracy level of 0.5683 and 0.5923, respectively, which is also very encouraging. Gaussian NB had an accuracy level of 0.5597, the lowest accuracy result of the seven machine algorithms classifiers used. Figure 1 gives the graphical representation of all 7 machine learning algorithms when plotted against the performance metrics.

Figure 1 gives a clear graphical representation of the Evaluation metrics and the algorithms using Dataset 1. The Bar graph represents each of the machine learning algorithms and the results they produced across each of the Evaluation metrics used.

The four ensemble learning models used in the course of this project, which include Ada Boost, Gradient Boosting, Random Forest, and Max Voting, were extensively trained using our datasets, and outputs were extracted from each of the algorithms.

**Table 4.** Result of Dataset 1 using the Machine Learning Algorithms and Ensemble learning methods.

| Model | Accuracy | Precision | Recall | Specificity | MCC | KAPPA | F1 Score | AUC | FDR | FNR | FPR | NVP |
|---|---|---|---|---|---|---|---|---|---|---|---|---|
| Decision Trees | 0.6487 | 0.6500 | 0.6500 | 0.6500 | 0.3000 | 0.3000 | 0.6500 | 0.6500 | 0.3500 | 0.3500 | 0.3500 | 0.6500 |
| The K-Nearest Neighbor | 0.5683 | 0.5700 | 0.5700 | 0.5700 | 0.1900 | 0.1200 | 0.5700 | 0.5600 | 0.4300 | 0.4300 | 0.4300 | 0.5700 |
| Logistic regression | 0.6923 | 0.6900 | 0.6900 | 0.6900 | 0.3900 | 0.3900 | 0.6900 | 0.6900 | 0.3100 | 0.3100 | 0.3100 | 0.6900 |
| Gaussian Naïve Bayes | 0.5597 | 0.5600 | 0.5600 | 0.5600 | 0.0800 | 0.0600 | 0.5600 | 0.5400 | 0.4400 | 0.4400 | 0.4400 | 0.5600 |
| Neural Network | 0.6827 | 0.6800 | 0.6800 | 0.6800 | 0.3600 | 0.3600 | 0.6800 | 0.6800 | 0.3200 | 0.3200 | 0.3200 | 0.6800 |
| Quadratic Discriminant Analysis | 0.5923 | 0.5900 | 0.5900 | 0.5900 | 0.2100 | 0.1900 | 0.5900 | 0.6000 | 0.4100 | 0.4100 | 0.4100 | 0.5900 |
| SVM | 0.6897 | 0.6900 | 0.6900 | 0.6900 | 0.3800 | 0.3800 | 0.6900 | 0.6900 | 0.3100 | 0.3100 | 0.3100 | 0.6900 |
| AdaBoost | 0.6227 | 0.6200 | 0.6200 | 0.6200 | 0.2900 | 0.2500 | 0.6200 | 0.6300 | 0.3800 | 0.3800 | 0.3800 | 0.6200 |
| Gradient Boosting | 0.6483 | 0.6500 | 0.6500 | 0.6500 | 0.3200 | 0.3000 | 0.6500 | 0.6500 | 0.3500 | 0.3500 | 0.3500 | 0.6500 |
| Random Forest | 0.6837 | 0.6800 | 0.6800 | 0.6800 | 0.3700 | 0.3700 | 0.6800 | 0.6800 | 0.3200 | 0.3200 | 0.3200 | 0.6800 |
| Max Voting | 0.7047 | 0.7000 | 0.7000 | 0.7000 | 0.4100 | 0.4100 | 0.7000 | 0.7000 | 0.3000 | 0.3000 | 0.3000 | 0.7000 |

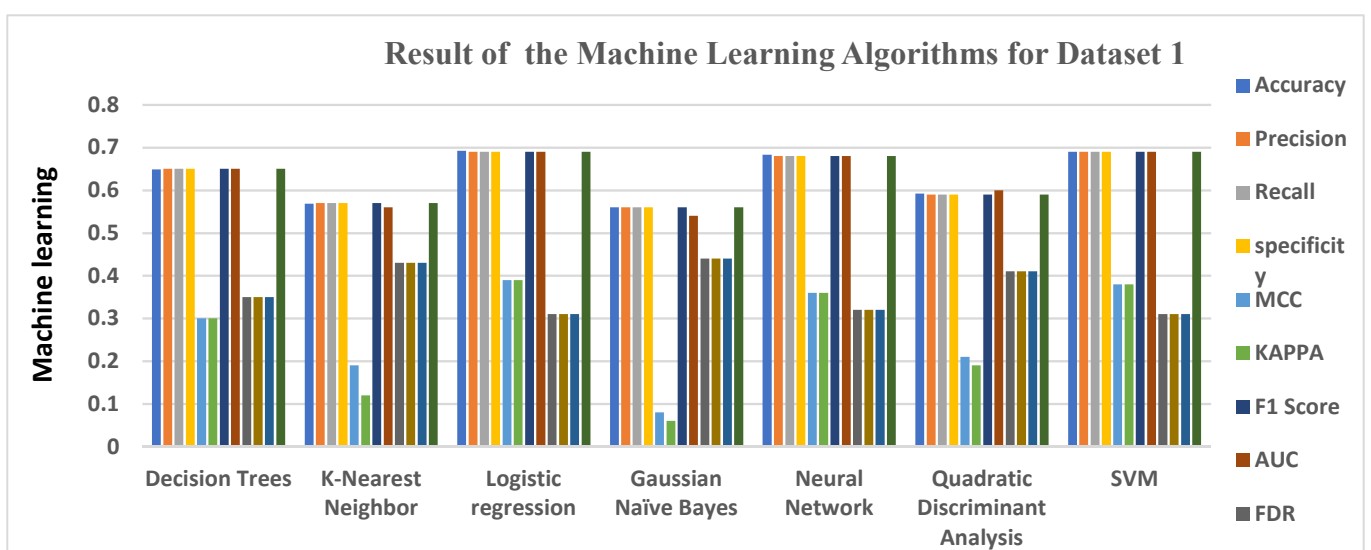

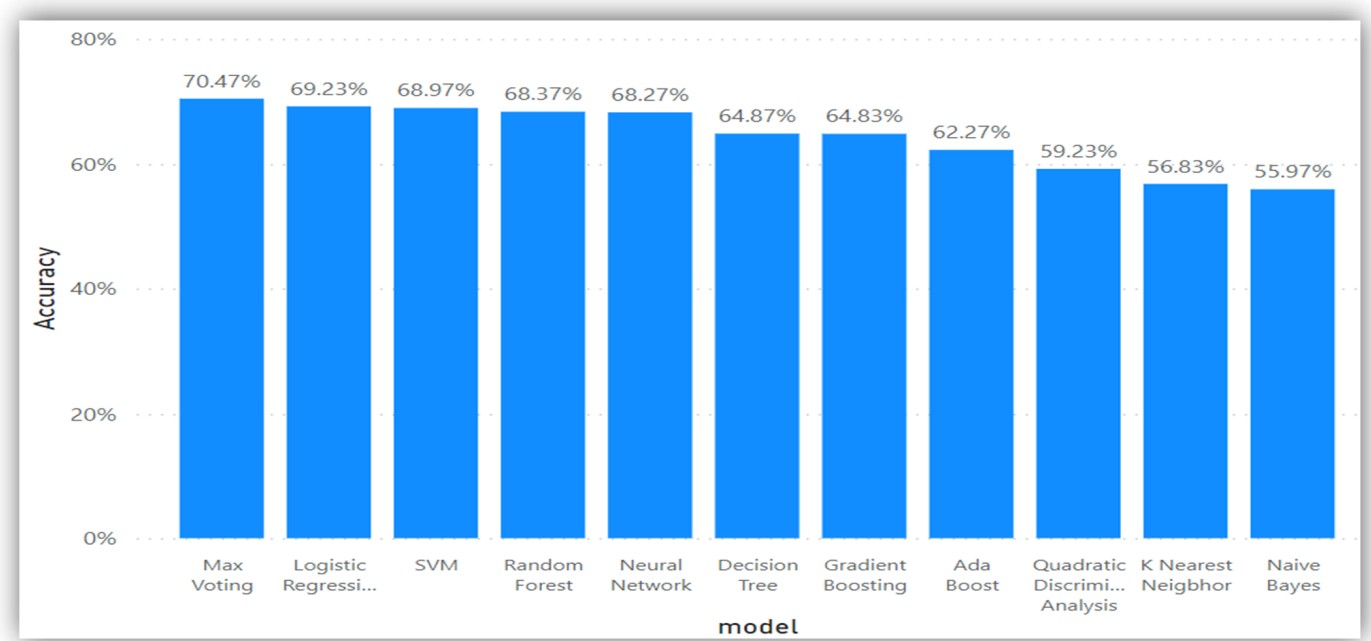

**Figure 1.** Graphical representations of Evaluation Metrics against Machine Learning Algorithms and ensemble learning for Dataset 1.

Each of the algorithms considering our performance metrics, as stated in chapter three, produced an output result consisting of Accuracy, Precision, Recall, Specificity, MCC, KAPPA, F1 Score, AUC, FDR, FNR, FPR, and NPV. The accuracy levels help us identify which of the classifiers performs best when it comes to detecting virtual harassment.

Considering the ensemble learning models on dataset 1, the result for the Ensemble classifiers on dataset 1 began with Ada Boost, which had the lowest accuracy level at 0.6227, followed by Gradient Boosting and Random Forest, which were at 0.6483 and 0.6837, respectively. The overall best ensemble performer for Dataset 1 was the Max Voting ensemble, which attained an accuracy of 0.7047. This is still the best in detecting a virtual harassment post in Dataset 1, although it produced the lowest FDR, FNR, and FPR results, at 0.30 (30%). However, Ada Boost had the lowest MCC and KAPPA of all the four ensemble learning methods, at 0.29 and 0.25, respectively, which is the sum of errors made for each example during the training and validation process.

Figure 2 gives a clear graphical representation of the Evaluation metrics and the algorithms using Dataset 1. The bar graph represents each of the ensemble learning models and the results they produced across each of the Evaluation metrics used.

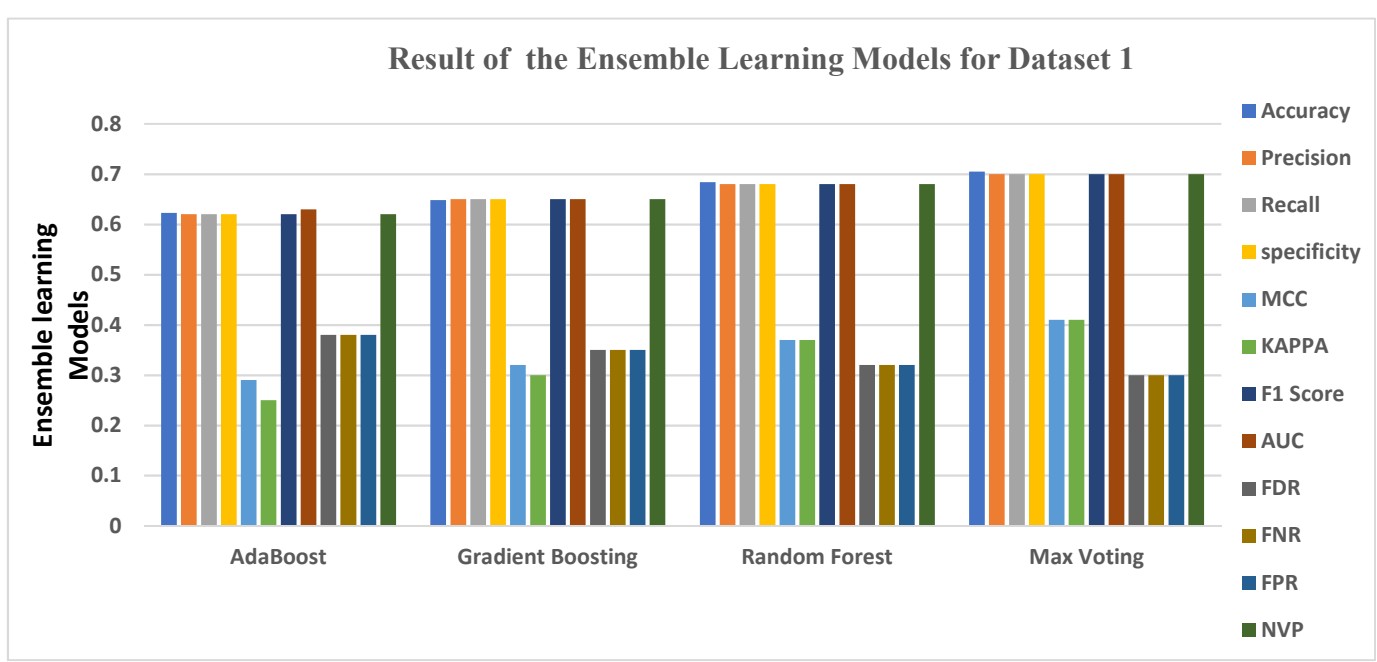

**Figure 2.** Graphical representation of Evaluation Metrics against Ensemble Learning Models for Dataset 1.

*5.2. Data Analysis for Dataset 2*

Table 5, using dataset 2, gives the result of 7 machine learning algorithms successfully trained using the dataset. Results were produced for each of the algorithms using the evaluation metrics stated earlier. Quadratic Discriminant Analysis QDA had the worst score of 0.1298. The Random Forest ensemble had the highest accuracy score of 0.8779, and KNN, Logistics Regression, and SVM all had accuracy scores of 0.8769. The same applied to the Gradient Boosting and Max Voting ensembles, both of which had an accuracy score of 0.8769.

Immediately following is the Neural Network, which had an accuracy level of 0.8731, followed by the Ada Boost Ensemble, which returned an accuracy score of 0.8654. This score is relatively high, considering the margin of the other three ensemble learnings, though it seems very low compared to the first three mentioned earlier (Random Forest, Max Voting, and Gradient Boosting). Decision Tree had an accuracy level of 0.8298, which is also high on the average accuracy score. Gaussian NB had the least accuracy level of

0.7981, which makes it suitable for detecting virtual harassment in comparison with the other accuracy score.

**Table 5.** Result of Dataset 2 using the Machine Learning Algorithms and Ensemble learning methods.

| Models | Accuracy | Precision | Recall | Specificity | MCC | KAPPA | F1 Score | AUC | FDR | FNR | FPR | NVP |
|---|---|---|---|---|---|---|---|---|---|---|---|---|
| Decision Trees | 0.8298 | 0.8383 | 0.8383 | 0.8383 | 0.1000 | 0.1000 | 0.8300 | 0.5400 | 0.1700 | 0.1700 | 0.1700 | 0.8300 |
| K-Nearest Neighbor | 0.8769 | 0.8800 | 0.8800 | 0.8800 | 0.0000 | 0.0000 | 0.8800 | 0.5000 | 0.1200 | 0.1200 | 0.1200 | 0.8800 |
| Logistic regression | 0.8769 | 0.8800 | 0.8800 | 0.8800 | 0.0000 | 0.0000 | 0.8800 | 0.5000 | 0.1200 | 0.1200 | 0.1200 | 0.8800 |
| Gaussian Naïve Bayes | 0.7981 | 0.8000 | 0.8000 | 0.8000 | 0.0100 | 0.0100 | 0.8000 | 0.5100 | 0.2000 | 0.2000 | 0.2000 | 0.8000 |
| Neural Network | 0.8731 | 0.8700 | 0.8700 | 0.8700 | 0.1100 | 0.0700 | 0.8700 | 0.5200 | 0.1300 | 0.1300 | 0.1300 | 0.8700 |
| Quadratic Discriminant Analysis | 0.1298 | 0.1300 | 0.1300 | 0.1300 | 0.0300 | 0.0000 | 0.1300 | 0.5000 | 0.8700 | 0.8700 | 0.8700 | 0.1300 |
| SVM | 0.8769 | 0.8800 | 0.8800 | 0.8800 | 0.0000 | 0.0000 | 0.8800 | 0.5000 | 0.1200 | 0.1200 | 0.1200 | 0.8800 |
| AdaBoost | 0.8654 | 0.8700 | 0.8700 | 0.8700 | 0.1800 | 0.1600 | 0.8700 | 0.5600 | 0.1300 | 0.1300 | 0.1300 | 0.8700 |
| Gradient Boosting | 0.8769 | 0.8800 | 0.8800 | 0.8800 | 0.1100 | 0.0600 | 0.8800 | 0.5200 | 0.1200 | 0.1200 | 0.1200 | 0.8800 |
| Random Forest | 0.8779 | 0.8800 | 0.8800 | 0.8800 | 0.0800 | 0.0100 | 0.8800 | 0.5000 | 0.1200 | 0.1200 | 0.1200 | 0.8800 |
| Max Voting | 0.8769 | 0.8800 | 0.8800 | 0.8800 | 0.0000 | 0.0000 | 0.8800 | 0.5000 | 0.1200 | 0.1200 | 0.1200 | 0.8800 |

In terms of other performance metrics, such as MCC and KAPPA, KNN, Logistics Regression, SVM, Max Voting, and QDA, all had the worst score of 0.0000. They all returned zero values. The Ada Boost, Gradient Boosting, Random Forest, and Max Voting ensembles all performed poorly at FDR, FNR, and FPR, respectively, as they produced results between 0.12 and 0.13.

Furthermore, as shown in the table, Random Forest, which has the highest accuracy, also produced a high precision score of about 0.8779. The same applied to its recall, specificity, F1 Score, and Negative Predictive Value (NPV). Random Forest had an average score for the AUC area of 0.50. The same average score applied to all other algorithms and ensemble learning, which all have an average between 0.50 and 0.56.

Figure 3 gives a clear graphical representation of the evaluation metrics, algorithms, and ensemble learning techniques using Dataset 2. The bar graph represents each of the machine learning algorithms, ensemble learning, and the accuracy results they produced across.

The four ensemble learning models used in the course of this project, which include Ada Boost, Gradient Boosting, Random Forest, and Max Voting, were extensively trained using our datasets, and outputs were extracted from each of the algorithms.

Each of the algorithms considering our performance metrics, as stated in chapter three, produced an output result consisting of Accuracy, Precision, Recall, Specificity, MCC, KAPPA, F1 Score, AUC, FDR, FNR, FPR, and NPV. The accuracy levels help us identify which of the classifiers performs best when it comes to detecting virtual harassment.

For the ensemble classifiers in Dataset 2, Ada Boost had the lowest accuracy level at 0.8654, followed by Gradient Boosting and Max Voting, which were both slightly higher by 0.0115, each achieving an accuracy score of 0.8769. Overall, the best ensemble performer for Dataset 2 is the Random Forest ensemble, which attained an accuracy of 0.8779, making it the most appropriate in detecting a virtual harassment post in Dataset 2, although it scored poorly in FDR, FNR, and FPR, at 0.12 (12%). This, however, was not as low as MCC and KAPPA, which both returned 0.00 (0%). However, Ada Boost has the highest MCC and KAPPA of all the four ensemble learning methods, at 0.18 and 0.16 respectively, which is the sum of errors made for each example during the training and validation process.

Figure 4 gives a clear graphical representation of the Evaluation metrics and the algorithms using Dataset 2. The bar graph represents each of the ensemble learning models and the results they produced across each of the evaluation metrics used.

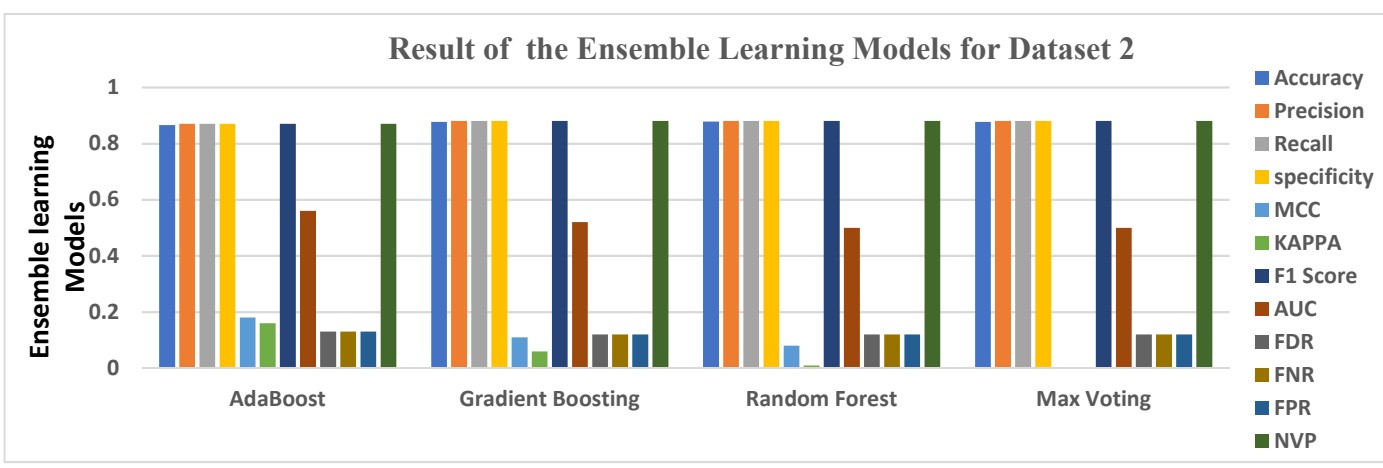

**Figure 3.** Graphical representations of Evaluation Metrics against Machine Learning Algorithms and ensemble learning for Dataset 2.

**Figure 4.** Graphical representation of Evaluation Metrics against Ensemble Learning Models for Dataset 2.

### 5.3. Data Analysis for Dataset 3

Table 6 gives the result of the 7 Machine algorithms and 4 ensemble learning using Dataset 3. The decision Tree attained an accuracy level of 0.9234. KNN had an accuracy of 0.9051. Logistics Regression also has an accuracy of 0.8896. SVM, QDA, and Neural Network also had accuracy results of 0.9243, 0.8528, and 0.9186, respectively. Gaussian NB, however, did not do so well in terms of accuracy, having an accuracy score of 0.8070, making it not so far from the true value but still within range.

**Table 6.** Result of Dataset 3 using the Machine Learning Algorithms and Ensemble learning methods.

| Model | Accuracy | Precision | Recall | Specificity | MCC | KAPPA | F1 Score | AUC | FDR | FNR | FPR | NVP |
|---|---|---|---|---|---|---|---|---|---|---|---|---|
| Decision Trees | 0.9234 | 0.9200 | 0.9200 | 0.9200 | 0.7600 | 0.7600 | 0.9200 | 0.8800 | 0.0800 | 0.0800 | 0.0800 | 0.9200 |
| K-Nearest Neighbor | 0.9051 | 0.9100 | 0.9100 | 0.9100 | 0.7000 | 0.7000 | 0.9100 | 0.8400 | 0.0900 | 0.0900 | 0.0900 | 0.9100 |
| Logistic regression | 0.8896 | 0.8900 | 0.8900 | 0.8900 | 0.6300 | 0.6000 | 0.8900 | 0.7600 | 0.1100 | 0.1100 | 0.1100 | 0.8900 |
| Gaussian Naïve Bayes | 0.8070 | 0.8100 | 0.8100 | 0.8100 | 0.6000 | 0.5500 | 0.8100 | 0.8600 | 0.1900 | 0.1900 | 0.1900 | 0.8100 |
| Neural Network | 0.9186 | 0.9200 | 0.9200 | 0.9200 | 0.7600 | 0.7500 | 0.9200 | 0.8800 | 0.0800 | 0.0800 | 0.0800 | 0.9200 |
| Quadratic Discriminant Analysis | 0.8528 | 0.8500 | 0.8500 | 0.8500 | 0.6700 | 0.6400 | 0.8500 | 0.8900 | 0.1500 | 0.1500 | 0.1500 | 0.8500 |
| SVM | 0.9243 | 0.9200 | 0.9200 | 0.9200 | 0.7600 | 0.7600 | 0.9200 | 0.8700 | 0.0800 | 0.0800 | 0.0800 | 0.9200 |
| AdaBoost | 0.8238 | 0.8200 | 0.8200 | 0.8200 | 0.3500 | 0.3200 | 0.8200 | 0.6300 | 0.1800 | 0.1800 | 0.1800 | 0.8200 |
| Gradient Boosting | 0.8266 | 0.8300 | 0.8300 | 0.8300 | 0.3500 | 0.2300 | 0.8300 | 0.5800 | 0.1700 | 0.1700 | 0.1700 | 0.8300 |
| Random Forest | 0.9258 | 0.9300 | 0.9300 | 0.9300 | 0.7700 | 0.7700 | 0.9300 | 0.8800 | 0.0700 | 0.0700 | 0.0700 | 0.9300 |
| Max Voting | 0.9254 | 0.9300 | 0.9300 | 0.9300 | 0.7700 | 0.7700 | 0.9300 | 0.8700 | 0.0700 | 0.0700 | 0.0700 | 0.9300 |

Table 6 also has other evaluation metrics results, such as Precision, Recall, Specification, F1 Score, and NPV, which we used across the following algorithms, ranging from Decision Trees, KNN, Logistics Regression, Gaussian NB, Neural Network, QDA, and SVM. All attained approximate scores of 0.92, 0.91, 0.89, 0.81, 0.92, 0.85, and 0.92 respectively.

The scores obtained for FDR, FNR, and FPR across all 7 algorithms ranged between 0.07 and 0.19, i.e., the range is between 7–19%. This result implies that only one evaluation metric cannot fully be enough to predict the efficiency of an algorithm, except tested with other metrics.

Figure 5 gives a clear graphical representation of the Evaluation metrics and the Algorithms using Dataset 3. The Bar graph represents each of the machine learning algorithms and the accuracy results they produced in percentages.

The four ensemble learning models used in the course of this project, which include Ada boost, Gradient Boosting, Random Forrest, and Max voting, as mentioned earlier, were extensively trained using our datasets. Outputs were collected from each of the algorithms. Each of the algorithms considering our performance metrics, as stated in chapter three, produced an output result consisting of Accuracy, Precision, Recall, Specificity, MCC, KAPPA, F1 Score, AUC, FDR, FNR, FPR, and NPV. The accuracy levels help us identify which of the ensemble classifiers performs best when it comes to detecting virtual harassment.

Table 6 also gives the result of the 4 ensemble learning models, highlighted in green. Random Forest had the highest accuracy level of 0.9258, Max Voting had an accuracy level of 0.9254, Gradient Boosting had 0.8266, and Ada Boost had the least accuracy level of 0.8238. While considering other evaluation metrics, Random Forest turns out to have a KAPPA score of 0.77 (77%), while Gradient Boosting had 0.23 (23%). All four ensemble learning obtained Precision, Recall, Specificity, F1 score, and NPV of 82%, 83%, 93%, and 93%, respectively, starting from Ada Boost, Gradient Boosting, Random Forest, and Max voting, making it very suitable for detecting virtual harassment in Dataset 3. QDA produced an AUC score of 0.89, making it the highest under AUC, showing perfection in the model's prediction.

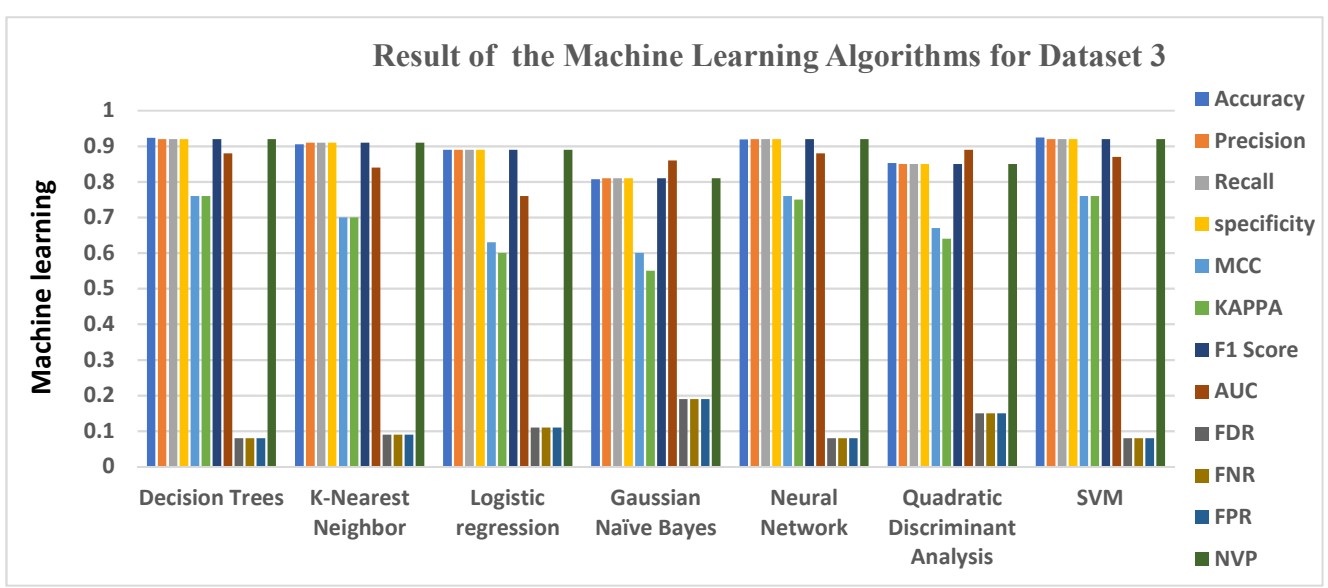

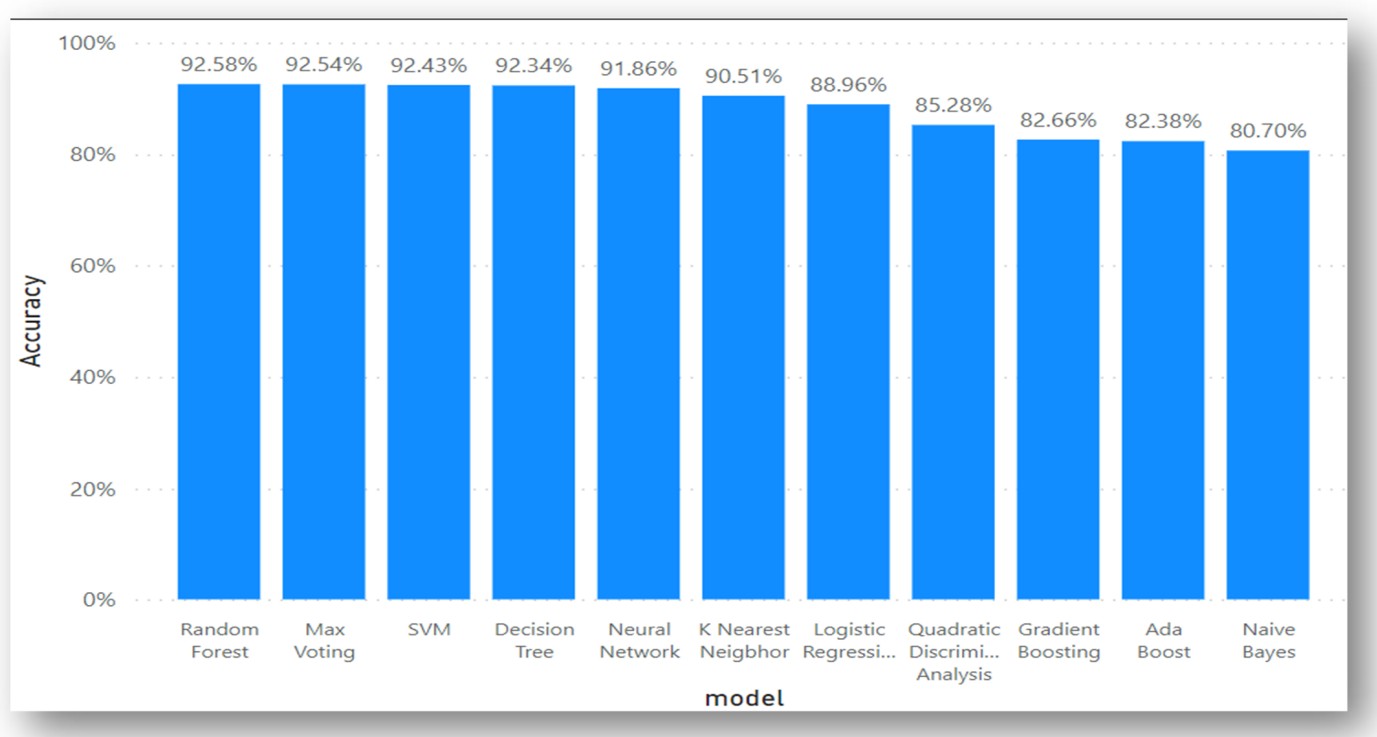

**Figure 5.** Graphical representations of Evaluation Metrics against Machine Learning Algorithms and ensemble learning for Dataset 3.

Figure 6 gives a clear graphical representation of the Evaluation metrics and the algorithms and the ensemble learning using Dataset 3. The Bar graph represents each of the ensemble learning models and the results they produced across each of the Evaluation metrics used.

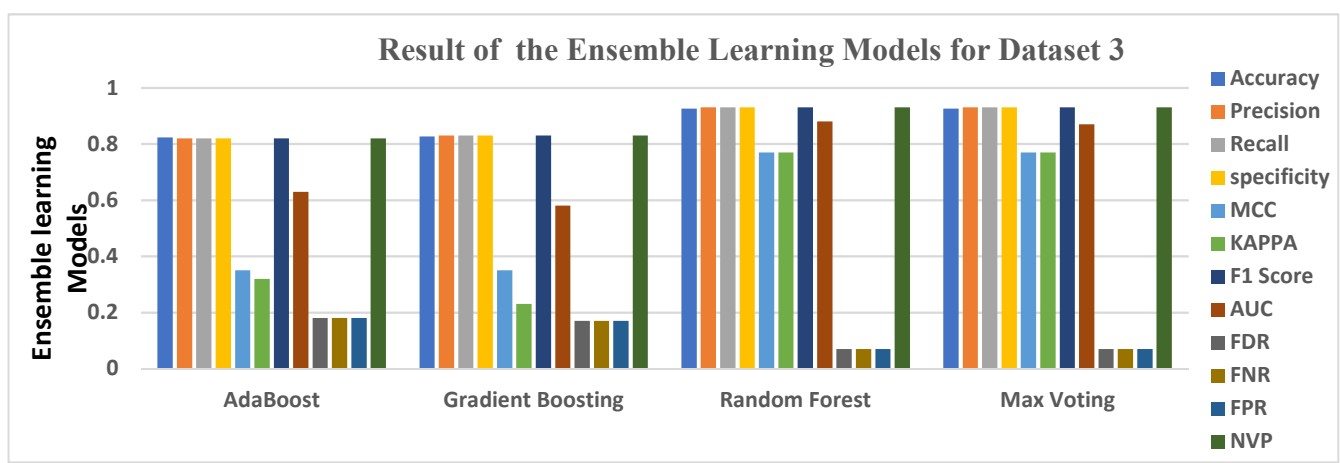

**Figure 6.** Graphical representation of Evaluation Metrics against Ensemble Learning Models for Dataset 3.

*5.4. Data Analysis for Dataset 4*

Table 7 gives the result of the 7 Machine algorithms as well as their ensemble learning counterpart using Dataset 4. Logistics Regression and SVM have the same highest accuracy level of 0.8383, respectively. Neural Network has an accuracy of 0.8083. Decision Tree had accuracy results of 0.7797, followed by Gaussian NB, with an accuracy of 0.6517, KNN and QDA also had accuracy results of 0.5577 and 0.5287, respectively, which is on the average level. The best matrices for Dataset 4 are Accuracy, Precision, Recall, Specificity, F1 score, and AUC. The second-best matrices are MCC and KAPPA, followed by the worst, which are FDR, FNR, and FPR.

**Table 7.** Result of Dataset 4 using the Machine Learning Algorithms and Ensemble learning methods.

| Model | Accuracy | Precision | Recall | Specificity | MCC | KAPPA | F1 Score | AUC | FDR | FNR | FPR | NVP |
|---|---|---|---|---|---|---|---|---|---|---|---|---|
| Decision Trees | 0.7797 | 0.7800 | 0.7800 | 0.7800 | 0.5600 | 0.5600 | 0.7800 | 0.7800 | 0.2200 | 0.2200 | 0.2200 | 0.7800 |
| K-Nearest Neighbor | 0.5577 | 0.5600 | 0.5600 | 0.5600 | 0.2100 | 0.1000 | 0.5600 | 0.5500 | 0.4400 | 0.4400 | 0.4400 | 0.5600 |
| Logistic regression | 0.8383 | 0.8400 | 0.8400 | 0.8400 | 0.6800 | 0.6800 | 0.8400 | 0.8400 | 0.1600 | 0.1600 | 0.1600 | 0.8400 |
| Gaussian Naïve Bayes | 0.6517 | 0.6500 | 0.6500 | 0.6500 | 0.3100 | 0.3100 | 0.6500 | 0.6500 | 0.3500 | 0.3500 | 0.3500 | 0.6500 |
| Neural Network | 0.8083 | 0.8100 | 0.8100 | 0.8100 | 0.6200 | 0.6200 | 0.8100 | 0.8100 | 0.1900 | 0.1900 | 0.1900 | 0.8100 |
| Quadratic Discriminant Analysis | 0.5287 | 0.5300 | 0.5300 | 0.5300 | 0.0900 | 0.0700 | 0.5300 | 0.5400 | 0.4700 | 0.4700 | 0.4700 | 0.5300 |
| SVM | 0.8383 | 0.8400 | 0.8400 | 0.8400 | 0.6800 | 0.6800 | 0.8400 | 0.8400 | 0.1600 | 0.1600 | 0.1600 | 0.8400 |
| AdaBoost | 0.7980 | 0.8000 | 0.8000 | 0.8000 | 0.6100 | 0.5900 | 0.8000 | 0.8000 | 0.2000 | 0.2000 | 0.2000 | 0.8000 |
| Gradient Boosting | 0.8010 | 0.8000 | 0.8000 | 0.8000 | 0.6200 | 0.6000 | 0.8000 | 0.8000 | 0.2000 | 0.2000 | 0.2000 | 0.8000 |
| Random Forest | 0.8210 | 0.8200 | 0.8200 | 0.8200 | 0.6500 | 0.6400 | 0.8200 | 0.8200 | 0.1800 | 0.1800 | 0.1800 | 0.8200 |
| Max Voting | 0.8280 | 0.8300 | 0.8300 | 0.8300 | 0.6600 | 0.6500 | 0.8300 | 0.8300 | 0.1700 | 0.1700 | 0.1700 | 0.8300 |

Table 7 also has other Evaluation metrics results; QDA on the other hand didn't do so well across the 12 performance matrices. We can see clearly from Table 7, that the QDA obtained an accuracy of 0.5287, precision, recall, specificity, F1 score, and NPV of 0.53 (53%) respectively, while FNR, FDR, and FPR all attained a score of 0.47 (47%), AUC had the best score of 0.54 (54%), MCC and KAPPA obtained the worst score at 0.09 (9%) and 0.07 (7%) respectively. This result implies that only one evaluation metric cannot fully be enough to predict the efficiency of an algorithm, except tested with other metrics.

Figure 7 gives a clear graphical representation of the Evaluation metrics and the Algorithms and ensemble learning using Dataset 4. The Bar graph represents each of the machine learning algorithms and the ensemble learning techniques and the results they produced.

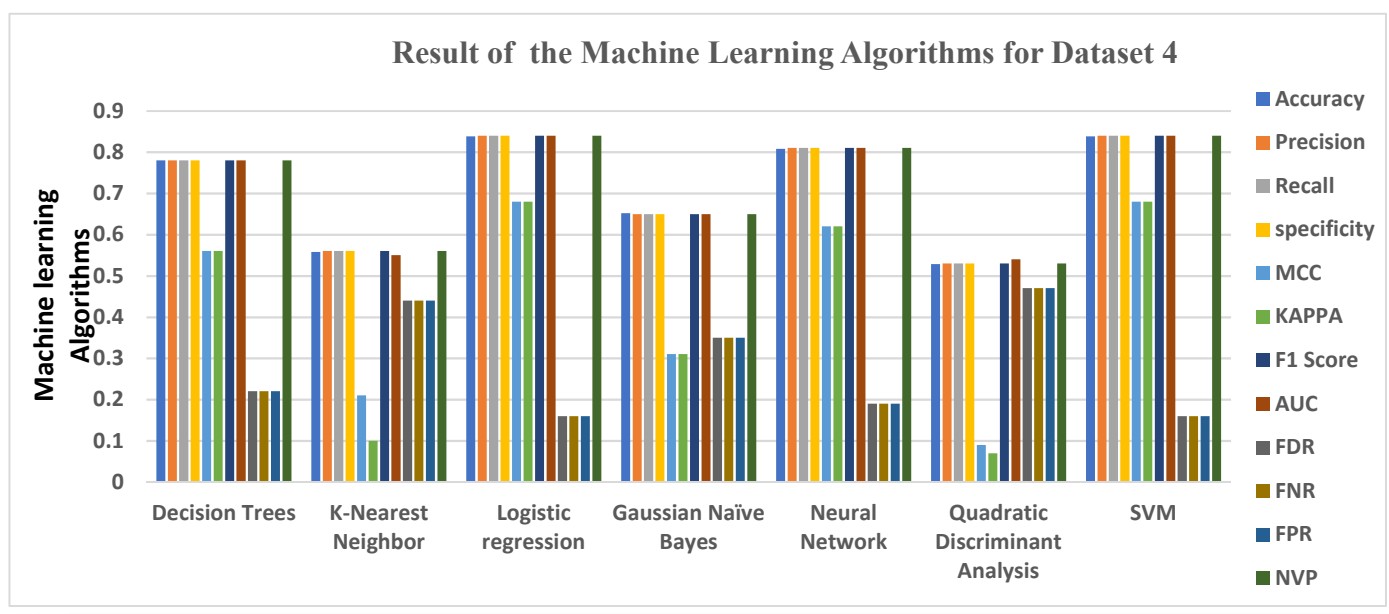

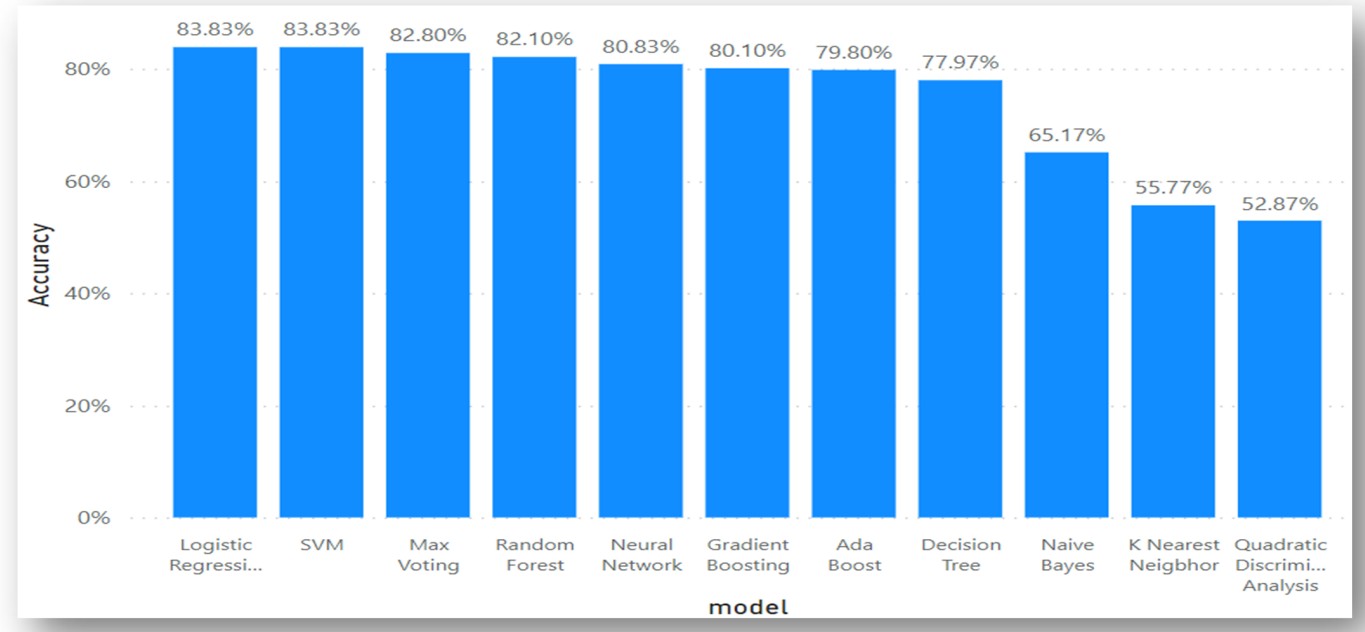

**Figure 7.** Graphical representations of Evaluation Metrics against Machine Learning Algorithms and ensemble learning for Dataset 4.

The four ensemble learning models—Ada boost, Gradient Boosting, Random Forest, and Max Voting—used in this study were thoroughly trained using our datasets, and results were obtained from each of the algorithms. Each algorithm that took into account the chapter three performance measures generated an output result that included Accuracy, Precision, Recall, Specificity, MCC, KAPPA, F1 Score, AUC, FDR, FNR, FPR, and NPV. The accuracy levels allow us to identify the ensemble classifiers that perform the best at identifying virtual harassment.

Table 7 also gives the result of the 4 ensemble learning models highlighted in green colour. Max Voting had the highest accuracy level of 0.8280, Random Forest had an accuracy level of 0.8210, Gradient Boosting had 0.8010, while Ada Boost had the least accuracy level of 0.7980. While considering other evaluation metrics, Ada Boost, Gradient Boosting, Random Forest, and Max voting all had Precisions, Recall, F1 Score, specificity, MCC, KAPPA, and NPV of above average, making them very suitable for detecting virtual

harassment in dataset 4, on the other hand, FDR, FNR, and FPR produced its low loss value of about 0.17 to 0.20, showing it is an imperfection in the model's prediction.

Figure 8 gives a clear graphical representation of the evaluation metrics and the algorithms using Dataset 4. The bar graph represents each of the ensemble learning models and the algorithms and the results they produced.

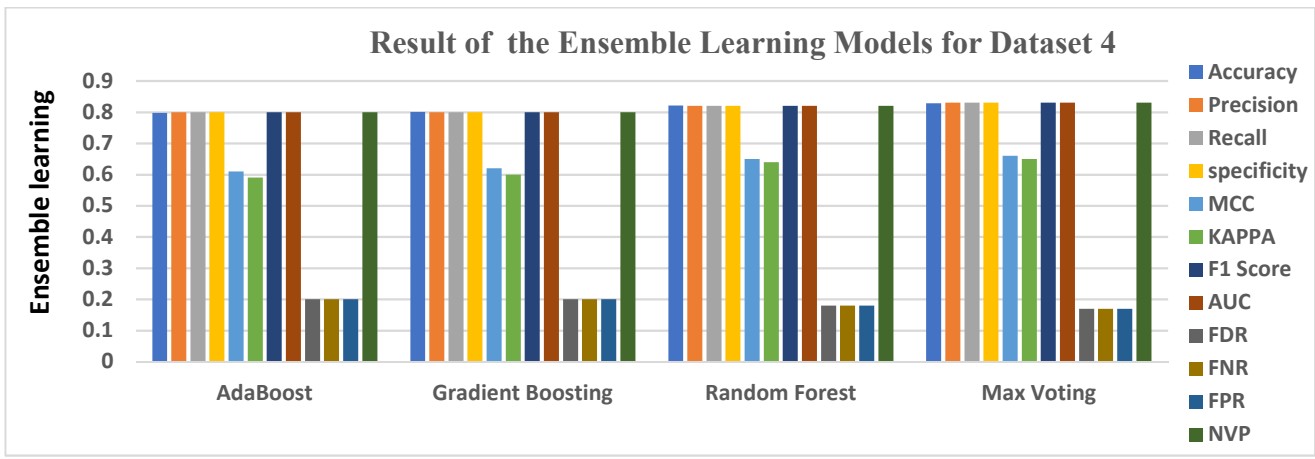

**Figure 8.** Graphical representation of Evaluation Metrics against Ensemble Learning Models for Dataset 4.

## 6. Future Work

In an attempt to enhance the performance of this work, the authors are planning to perform similar experiments on more datasets that share the same characteristics and features as those considered. Apart from this, the intention to further this work is also premised on the increase in the number of algorithms adopted with the aim of broadening the scope and obtaining better performance with the introduction of hyperparameter optimization. Consequently, the implementation of this work in a real-life scenario is of the utmost importance.

## 7. Conclusions

Virtual harassment is considered to be one of the biggest issues with the development of technology (Internet), social media, and various online communications. It can be carried out by a single user or a group of users who use the internet to harass, embarrass, afflict, torment, and make a nuisance of a specific person online, which has caused serious health problems and is still causing serious health problems to this day, including suicide, depression, and other mental health issues.

To train our machine learning classifiers for classifying comments as virtual harassment or non-virtual harassment, a virtual harassment model was developed to detect virtual harassment comments across four different datasets while taking into account the users' features, activity features, and content features. Our machine learning methods, including Decision Tree, K Nearest Neighbor, Logistic Regression, Gaussian NB, Neural Network, Quadratic Discriminant Analysis, and Support Vector Machine, were trained through extensive experiments. Additionally, experiments were conducted utilizing the Ada Boost, Gradient Boosting, Random Forest, and Max Voting ensemble learning models. After utilizing the datasets to train our algorithms, the algorithms were tested and trained using the datasets, the results for the accuracy, precision, recall, specificity, F1 Score, MCC, KAPPA, FDR, FNR, FPR, AUC, and NPV were obtained. Detailed results are shown in Tables 1–7 for the machine learning and ensemble learning models and how they performed across various datasets. Overall, the Bayzick Dataset 3 [77] performed best out of the four datasets used and the worse metrics were the FDR, FNR, and FPR out of the twelve metrics used.

The ensemble learning models outperformed the machine learning models in the evaluation measure produced after the tests because they had access to more data to learn from, as opposed to the machine learning algorithms. When the machine learning algorithms' assessment metrics are contrasted with those for the ensemble learning models, as given in Tables 1–7, (in terms of Accuracy, Precision, Recall, Specificity, F1 Score, MCC, KAPPA, FDR, FNR, FPR, AUC, and NPV). Feature engineering is not necessary for ensemble learning models because they are capable of carrying out feature engineering on their own by scanning the dataset for correlated features and combining them for quick learning without being explicitly told to. Although machine learning performs better with small datasets, ensemble learning models need more data to fully realize their potential. As a result, the potential of ensemble learning models is not fully realized with short datasets.

**Author Contributions:** Methodology, N.A.A. and E.F.; software, N.A.A. and E.F.; formal analysis, N.A.A. and E.F.; writing—original draft, N.A.A.; writing—review & editing, N.A.A. and E.F. All authors have read and agreed to the published version of the manuscript.

**Funding:** This work was supported by the Deanship of Scientific Research, Vice Presidency for Graduate Studies and Scientific Research, King Faisal University, Saudi Arabia [Grant No. 2526].

**Data Availability Statement:** No data were used to support the findings of the study.

**Acknowledgments:** The author would like to thank Scientific Research, Vice Presidency for Graduate Studies and Scientific Research, King Faisal University, Saudi Arabia for support.

**Conflicts of Interest:** The authors declare no conflict of interest.

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
