# Peer review of "Classification of Virtual Harassment on Social Networks Using Ensemble Learning Techniques"

_applsci, doi:10.3390/app13074570_

Round 1
Reviewer 1 Report
The theme is very interesting and updated for the scientific community. The paper is well structured. This study has a good background study to support the results. The findings are in line with the results obtained.
Author Response
Kindly see attached document.

Reviewer 2 Report
This paper focuses on the effect of traditional classifiers and ensemble learning on the classification of virtual harassment in online social media networks under different data sets training, which can effectively identify virtual harassment in the network and purify the network environment. This research direction has positive significance for the spirit and psychology of online social media users. However, the paper needs to make some modifications before receiving the publication. My detailed comments are as follows :
1. The method section in the abstract mentions four ensemble learning models, why only three are listed ?
2. How is the number of virtual harassment counted in Figure 2 ? Is it the sum of all social software or statistics for one software ?
3. There are many definitions of virtual harassment in the second section(Materials and Methods), but their intrinsic meanings are similar, so they can be modified.
4. Please pay attention to the order of references, they should be sorted from small to large, so that the article will look more neat.
5. Some sentences need to be written concisely so that readers can easily understand them. For example, 'used for an online toxic comment classification challenge used to identification and classification of online toxic comments. '
6. In this paper, whether some professional terms are explained, such as the definition of confusion matrix and the meaning of performance parameters.
7. The fourth chapter has multiple repeated paragraphs, please check whether these paragraphs need to be differentiated.
8. Which graph does Figure 4.9 in Section 4.2 refer to ? Please use the correct picture serial number.
9. Two papers about network should be cited:
[1] Traffic dynamics on multilayer networks with different speeds. IEEE Transactions on Circuits and Systems II: Express Briefs. 2022, 69(3): 1697 - 1701.
[2] An improved optimal routing strategy on scale-free networks. IEEE Transactions on Circuits and Systems II: Express Briefs. 2022, 69(11): 4578 - 4582.
Author Response
Kindly see the attached document.

Reviewer 3 Report
1. I suspect the abstract is quite long. Please, summarize in some points. Moreover, the second sentence of the abstract is too long (split it) and the the keywords background, methods, results should be put in italic font.
2. Introduction needs much work. Please, highlight here in different paragraphs, perhaps styling like the abstract with a keyword in italic before them, the following key elements: i) the scenario, ii) the problem faced, iii) the motivation and deficiency in the literature, iv) the objectives / the research questions, v) the proposed approach in brief, vi) the major contributions and the gaps of the literature filled by your work, as well as the key novelties, vii) the structure of the paper with links to the sections.
3. Section 2 actually starts with a rationale that provides clues form the scenario of online social media/network and virtual harassment. Please move this in part to the introduction. Section 2 should be a Background section that includes a rationale of the proposal where authors should discuss the graphics and the statistics already shown and the basic information about the methods and materials used.
4. All references [x] should be put before the full stop at the end of the periods, not after. There are many cases like the one at lines 79-80. Please, perform proof-reading and grammar check. Moreover, at line 171, as well as others, there is an invalid reference reported in APA format instead of the Journal's template.
5. Lines 122-127 have some errors. The same applies at line 181 and 166.
6. The related work as it is now is mainly a listing. Authors should compare point-to-point to each work just after they describe it. In the table 1 authros should show different criteria for the comparison and put the current paper at the first or last line, highlighted, to ease the reading of anyone interested so that he/she can grasp intuitively the novelties/differences/similarities.
7. Authors need a methodology section with an overview of the steps and then describe over different subsections the various steps.
8. Please ground the approach to combine classifier to existing and recent literature (see https://doi.org/10.1371/journal.pone.0245230, https://doi.org/10.1007/s00521-022-07454-4, https://doi.org/10.1007/s11042-020-10446-y, https://doi.org/10.1016/j.inffus.2017.02.007, https://doi.org/10.1007/s00521-013-1362-6). Why not using the other techniques shown in the previous papers? Please motivate the choice.
9. Please, perform statistical tests to find the best solution.
10. Did authors perform parameter tuning? If so, which parameters have been tuned for each classifier used? And what are the best hyper-parameter? Which procedure has been used for tuning?
11. Add limitations, expected impact and future works in the conclusion part.
12. Can authors share the code used? It can be put in the appendix or on a Github repository.
13. What about systems like the ones at https://doi.org/10.1016/j.comnet.2021.108614 or Wang, Q., Xue, H., Li, F., Lee, D., & Luo, B. (2019). # DontTweetThis: Scoring private information in social networks. Proceedings on Privacy Enhancing Technologies, 2019(4), 72-92. or patent like https://patents.google.com/patent/US20150365366. Can be used (when needed, properly tweaked) to mitigate the virtual harassment problem? Please, discuss about this.
Author Response
Kindly see the attached document.

Round 2
Reviewer 2 Report
In the new version manuscript, the paper has been great improved. All my concerns have been properly addressed. Thus, I recommend the acceptance of this paper.
Reviewer 3 Report
The authors have replied to all my comments.
Good job!
Please, correct the "key error" on the GitHub page.